# A Lightweight AlCrTiV_0.5_Cu*_x_* High-Entropy Alloy with Excellent Corrosion Resistance

**DOI:** 10.3390/ma16072922

**Published:** 2023-04-06

**Authors:** Zhen Peng, Baowei Li, Zaibin Luo, Xuefei Chen, Yao Tang, Guannan Yang, Pan Gong

**Affiliations:** 1School of Materials Science and Engineering, Jiangsu University, Zhenjiang 212013, China; 2China Steel Development Research Institute, No.95 Longfusi Street, Dongcheng District, Beijing 100009, China; 3State Key Laboratory of Precision Electronic Manufacturing Technology and Equipment, Guangdong University of Technology, Guangzhou 510006, China; 4State Key Laboratory of Materials Processing and Die & Mould Technology, School of Materials Science and Engineering, Huazhong University of Science and Technology, No. 1037 Luoyu Road, Wuhan 430074, China; pangong@hust.edu.cn

**Keywords:** corrosion resistance, high entropy alloys, microstructure, lightweight

## Abstract

Lightweight high-entropy alloys (HEAs) are a new class of low-density, high strength-to-weight ratio metallic structural material. Understanding their corrosion behavior is crucial for designing microstructures for their practical applications. This work investigates the electrochemical corrosion behavior of lightweight HEAs AlCrTiV_0.5_Cu*_x_* (*x* = 0, 0.2, 0.4, 0.6, 0.8, and 1.0) in a 0.6 M NaCl solution. These HEAs were produced by vacuum arc melting. In contrast to 304L stainless steel, all of the alloys exhibited lower current density levels caused by self-corrosion, with AlCrTiV_0.5_ demonstrating the highest corrosion resistance (0.131 μA/cm^2^). Corrosion resistance decreased along with the content of copper because copper segregation accelerated local corrosion throughout the alloy.

## 1. Introduction

High-entropy alloys, also known as HEAs, are generally composed of five or more elements in equimolar or near-equimolar ratios. These alloys are stable in a disordered solid solution state owing to a high mixing entropy [1,2]. Over the past twenty years, there has been an explosion of interest in a novel concept of alloy design that allows for the creation of materials with superior mechanical or physical properties. These properties include high yield strengths, ductility, and fracture toughness, in addition to superior high-temperature oxidation resistance and corrosion resistance. As a result of these features, HEAs have found applications in a huge range of different fields [3,4,5,6,7,8,9,10,11,12,13].

However, despite the application potential being high, the high density of many HEAs greatly inhibits the actual deployment of these applications. For example, the density of a CoCrFeNiMn HEA is 8.1 g/cm^3^, while the density of refractory high-enthalpy alloys (HEAs) is more than 10 g/cm^3^. Both of these HEAs have a mass density that is higher than that of steel, which has a density of 7.9 g per cubic centimeter. For the purpose of increasing the effectiveness of systems that convert energy, it is necessary to reduce the weight. This necessitates the use of building and engineering materials that have a low density [14,15,16,17,18,19,20]. Recent studies have shown a significant interest in the development of high-entropy alloys that are also lightweight. While making lightweight HEAs, it is common practice to incorporate significant amounts of light components such as aluminium, titanium, magnesium, and lithium. The incorporation of Al (also for the purpose of alloy strengthening) changes the phase composition and microstructure of HEAs to produce a multiphase microstructure that can include FCC, BCC, and B2 structures [21,22], and the incorporation of Ti can easily lead to the production of secondary phases. Single-phase equiatomic AlCrTiV HEA has a low density of 5.06 g/cm^3^ and a hardness of 498 HV. The AlCrTiV quaternary system was selected by Huang et al. [23], who further increased the material’s hardness by including lightweight microalloying components. Microalloyed AlCrTiV alloys have a maximum hardness of 710 HV and a density of around 4.5 g per cubic centimeter. In addition to their mechanical advantages, high-performance additives provide excellent resistance to corrosion, which is an important factor in the application of engineering and structural materials.

The microstructure, alloying components, and production procedure of lightweight HEAs are the primary factors that decide the corrosion behavior in general. Tseng and colleagues [24] have developed a lightweight HEA material with the formula Al_20_Be_20_Fe_10_Si_15_Ti_35_ that has a low density of 3.91 g/cm^3^ and a high hardness of 911 HV. At temperatures of 700 and 900 °C, the alloy had an oxidation resistance that was remarkable and was much superior to that of Ti6Al4V. Ji et al. [25] conducted a study in which they manufactured a low-cost Al_35_Mg_30-*x*_Zn_30_Cu_5_Si*_x_* HEA that had a low density. They found that the resistance of as-cast alloys to corrosion in a solution containing 3.5 wt% NaCl may be improved by increasing the Si/Mg ratio. They also found that corrosion occurred primarily at the grain boundaries and gradually spread into the eutectic and intermetallic phases that included magnesium. This was a new finding for them. According to the findings of Qiu et al. [26], AlTiVCr HEA exhibited remarkable resistance to corrosion in 0.6 M NaCl, much higher than that of pure Al and 304SS.

It is possible to improve the qualities of a material by designing the formation of secondary phases and intermetallic phases inside the substance. The same strategy for strengthening materials, namely alloying, was used in the development of HEAs. The addition of Cu to HEAs is a practical method that may be used to achieve improved properties [15,27]. Yet, when exposed to aqueous environments, the heterogeneous microstructures of HEAs have the potential to cause localized corrosion. It is very necessary to investigate the impact that Cu concentration has on the corrosion resistance of HEAs in order to guarantee the consistency of industrial applications. In this study, lightweight HEAs made of AlCrTiV_0.5_Cu*_x_* were created by vacuum arc melting. Their electrochemical corrosion behavior in 0.6 M NaCl solution, composition homogeneity, and multi-scale microstructure were investigated.

## 2. Experimental Materials and Methods

### 2.1. Material Preparation

The AlCrTiV_0.5_Cu*_x_* lightweight HEAs were manufactured in a graphite crucible by the process of vacuum arc melting under the protection of an Ar atmosphere. We picked raw minerals that had an extremely high level of purity, such as aluminum (99.5%), chromium (99.95%), titanium (99.95%), vanadium (99.95%), and copper (99.95%). Each sample was remelted at least six times in an ingot to ensure that the chemical composition remained consistent throughout the melting process and to decrease the amount of oxidation that occurred. A specimen measuring 5 mm × 5 mm × 5 mm was extracted from the selected area in the middle of the ingot by using a wire cutter that was coupled to an electric discharge machine. Every sample was given a grit-2000 grinding on SiC sandpaper before being subjected to ultrasonic cleaning in ethanol.

### 2.2. Microstructural Characterization

In order to determine the phase structure of the samples, a Rigaku SMARTLAB9 X-ray diffractometer was used with Cu-Kα radiation across a range of 10° to 90° and at a scanning rate of 10°/min. The microstructure and corrosion morphologies were investigated using scanning electron microscopy (SEM) and X-ray energy dispersive spectroscopy (EDS), respectively.

### 2.3. Electrochemical Corrosion Measurements

Electrochemical corrosion tests were carried out at a temperature of 25 °C in 0.6 M NaCl in the presence of air. In order to carry out these measurements, a diamond polishing paste with particles of 1.5 μm in size was applied to specimens that had a cross-sectional area of 0.25 cm^2^. Using a CHI760 electrochemical workstation that was supplied with a standard three-electrode setup, polarization curves and electrochemical impedance spectra (EIS) were measured. The corroded samples were used as working electrodes, a AgCl electrode was used as the reference electrode, and a sheet of platinum was used as the counter electrode in this experiment. The open circuit potential (OCP) was monitored for a period of thirty minutes prior to the EIS and polarization studies in order to guarantee a consistent potential for the duration of the tests. The EIS measurements were collected with a potential amplitude of 5 mV and with frequencies ranging from 100 kHz to 10 MHz. Potentiodynamic polarization curves were obtained with a scan rate of 3 mV/s, with a starting potential of −0.9 V_SHE_ and an ending potential of 1.1 V_SHE_. In order to guarantee the accuracy of the results, more than three measurements were obtained for each different test situation. When the potentiodynamic polarization studies were finished, the samples were wiped clean with ethanol and then left to dry in the air. When the electrochemical tests were completed, the surface morphology of the samples was analyzed using a field emission scanning electron microscope (FEI Nova Nano450) (FESEM).

## 3. Results and Discussion

### 3.1. Microstructure Characterization

Cu has a positive binary enthalpy of mixing and relatively high valence electron concentration (VEC) with other constitutional elements. Figure 1 shows that the mixing enthalpy, VE, and mixing entropy Δ*S*_mix_ of the AlCrTiV_0.5_Cu*_x_* HEAs increased with the increase in Cu content, while the atomic size difference *δ* decreased with Cu addition. Hence, the microstructure and phase stability of AlCrTiV_0.5_Cu*_x_* HEAs may vary with the composition. Cu tends to segregate out of the matrix due to a positive binary enthalpy of mixing of Cu with other constitutional elements. The AlCrTiV_0.5_Cu*_x_* HEAs have a relatively large atomic size difference. Based on the solid-solution phase formation rules of high entropy alloys [28], there are probably ordered phases such as intermetallic compounds in the matrix. According to the VEC criterion [29], the phase composition of AlCrTiV_0.5_ is mainly BCC phase. The content of FCC phase increases with the addition of Cu, and AlCrTiV_0.5_Cu is mainly FCC phase. Figure 2 shows XRD patterns of AlCrTiV_0.5_Cu*_x_* samples. In summary, with the addition of Cu, the phase composition of AlCrTiV_0.5_Cu*_x_* HEAs may gradually change from BCC phase to FCC phase, and there may be intermetallic compounds. The segregation of Cu between grains increases with the copper content.

Figure 3 shows the SEM images of the AlCrTiV_0.5_Cu_x_ lightweight HEAs. AlCrTiV_0.5_ exhibits a typical equiaxed crystal structure with uniform composition and without obvious precipitation phases. With the addition of Cu, the alloys AlCrTiV_0.5_Cu_0.2_, AlCrTiV_0.5_Cu_0.4_, AlCrTiV_0.5_Cu_0.6_, and AlCrTiV_0.5_Cu_0.8_ show an equiaxed dendritic structure with more obvious grain boundaries. Moreover, the grain size gradually decreased with increasing Cu content, When the Cu content is further increased, AlCrTiV_0.5_Cu shows a non-equiaxial dendritic structure with a further reduction in grain size. We presented the X-ray diffraction pattern of lightweight AlCrTiV_0.5_Cu*_x_* HEAs in a previous work [15].

Further in-depth research on the elemental composition and dispersion of lightweight AlCrTiV_0.5_Cu*_x_* HEAs was conducted. The EDS maps of an enlarged portion of the samples are shown in Figure 4 (which shows both the grains and grain borders). The findings indicate that the components are dispersed throughout the samples in an even manner on a microscopic scale. Since Cu is missing, there is no segregation of the elements, and they are all distributed in the same manner. Nevertheless, the results of the XRD investigation reveal that the segregation gets more prominent with increasing Cu concentration. This results in the development of an Al-Ti-Cu-rich HCP phase as well as a V-Cr-rich FCC phase. The inclusion of copper is largely responsible for this kind of alloy having the dendritic structure that is so distinctive in other alloys of this type.

### 3.2. Electrochemical Corrosion Behavior in NaCl solution

The potentiodynamic polarization curves of the AlCrTiV_0.5_Cu*_x_* lightweight HEAs in 0.6 M NaCl solution are shown in Figure 5 and the results of the calculations used to determine the electrochemical corrosion parameters are shown in Table 1. The open-circuit potential of a material is referred to as its corrosion potential, or *E_corr._* It is possible to determine the corrosion rates of materials by utilizing the corrosion current density (*I_corr_*). In order to calculate *I_corr_* from the Tafel diagram, the linear portion of the polarization curve located close to *E_corr._* is extrapolated. It is clear from both the fitting parameters and the potentiodynamic polarization curves of the alloys that, as the concentration of copper increases, the corrosion current density (*I_corr_*) also increases, but the resistance to corrosion decreases. AlCrTiV_0.5_ (0.131 A/cm^2^) had the lowest value of *I_corr_* and the maximum corrosion resistance out of the six alloys that were tested. This is because the addition of Cu to the alloy will cause segregation at the grain boundaries. Dendrites may be produced by combining a Cu-depleted region with a Cu-rich region. If there is a significant potential difference between the two phases, galvanic coupling can result in corrosion, with the interdendritic area corroding first and increasing the local corrosion of the alloy [30,31]. If there is a significant potential difference between the two phases, galvanic coupling can also result in corrosion. Because of this, the corrosion resistance of the AlCrTiV_0.5_Cu*_x_* lightweight HEAs in NaCl solution gradually decreases as the content of Cu increases.

Figure 6 displays the Nyquist, Bode, and phase-angle charts of the lightweight AlCrTiV_0.5_Cu*_x_* HEAs at a range of temperatures corresponding to varied operating conditions. Z-view was used in order to evaluate the fitted parameters, and Figure 6 presents the equivalent circuit that was produced as a consequence.

A larger semicircle radius indicates that the interface has a higher resistance for the charge transfer and is a more protective passive film [32,33,34,35]. Figure 6a depicts the Nyquist plot, and it has the shape of an incomplete semicircle, which indicates that the charge transfer process is in control of the corrosion process. In the high-frequency range, incomplete capacitive arcs are shown by the AlCrTiV_0.5_Cu*_x_* lightweight HEAs. This is evidenced by Figure 6a, which makes it abundantly clear that the incomplete capacitance arcs are caused by charge transfer at inhomogeneous surfaces. As the concentration of copper in the alloy increases, its resistance to corrosion diminishes, and the radius of the capacitive semicircle becomes smaller.

Figure 6b,c show the Bode and phase angle charts of the alloys’ electrochemical corrosion, respectively. As can be seen, the impedance modulus and phase-angle change with frequency. The Bode plot shown in Figure 6b demonstrates that the passive films have a pseudocapacitive character since the slopes are less than −1 and the phase angle is smaller than −90°. In Figure 6c, the phase angle is getting close to 90°, and the value of the impedance modulus is linear from 1 Hz all the way up to 103 Hz. The phase angle of the AlCrTiV_0.5_Cu*_x_* alloys decreases from 103 Hz, achieves a tiny value at 104 Hz, and then continues to rise and reaches a high value at 105 Hz. A low value is reached again at 10^4^ Hz, and a high value again at 10^5^ Hz.

The data on the impedance are used in the production of an equivalent electric circuit (EEC) (shown in Figure 7). R1 represents the resistance of the solution in the EEC model, R2 represents the resistance of the passivated layer, and R3 represents the charge transfer resistance of the limited corrosion zone. In the zone that has been passivated, the capacitance is represented by CPE1, and in the zone that has been partly corroded, it is represented by CPE2. A constant phase-angle element, also known as a CPE, is used in place of a traditional capacitor so that flaws in capacitive components, such as surface inhomogeneities, may be taken into consideration. If the value of the charge transfer resistance, or R3, is lowered, then a greater number of electrons and ions will be able to flow through the passivation layer created by the alloy, which will result in a reduction in the alloy’s resistance to corrosion [36,37,38,39].

Table 2 presents some example parameters for EEC fitting that were generated while operating under OCP conditions. When the value of CPE1-P is close to 0.9, this indicates that the properties of the component are between those of a typical capacitor and those of a Warburg impedance [40,41,42,43,44]. The charge transfer resistance, denoted by R3, will decrease in proportion to the amount of copper that is present. We discover that AlCrTiV_0.5_ has a high charge transfer resistance of 437,920 Ω·cm^2^ (and strong corrosion resistance in 0.6 M NaCl solution), while AlCrTiV_0.5_Cu possesses the lowest charge transfer resistance of 31,276 Ω·cm^2^. According to the data presented above, the corrosion process of AlCrTiV_0.5_Cu*_x_* HEAs is regulated not only by the transfer of charge but also by the regulation of diffusion.

### 3.3. Corrosion Morphology Analysis

Figure 8 depicts the surface morphology of the AlCrTiV_0.5_Cu*_x_* lightweight HEAs after they were subjected to polarization tests in a solution containing 0.6 M NaCl. The corrosion has reached such an advanced stage that the surfaces are severely exfoliated and are in no way uniform. The fact that there are just a few pits on the surface is evidence of limited corrosion; hence, the samples were not fully ruined. The surface roughness was not on the nanoscale scale in the regions of the material that did not include corrosion pits.

Figure 8a depicts the surface morphology of the lightweight AlCrTiV_0.5_ HEA, which demonstrates that the surface is smooth and flat after corrosion and does not exhibit any obvious corrosion pits. The prior results that the AlCrTiV_0.5_ alloy exhibited greater corrosion resistance than the other five HEAs are supported by these new data, which are consistent with those findings. AlCrTiV_0.5_ lightweight HEA has high resistance to corrosion as a result of its uniform composition, single BCC phase structure, and lack of composition segregation at grain boundaries. These characteristics contribute to the material’s outstanding uniformity. When AlCrTiV_0.5_Cu*_x_* is corroded (where *x* may be any of the values 0.2, 0.4, 0.6, 0.8, or 1.0), substantial aggregation of Cu elements occurs in the dendritic intergranular region (see Figure 8), and enormous corrosion pits also develop. When there is a higher percentage of copper in the alloy, the surface corrosion of the alloy becomes worse.

In general, the AlCrTiV_0.5_ lightweight HEAs show good corrosion resistance in the testing environment. Increased Cu content enhanced Cu segregation, which could induce localized corrosion susceptibility in Cu-added HEAs. The addition of Cu also increases the mixing entropy of the system, thus the effect of the high-entropy effect on the corrosion resistance of the alloys should be considered. The corrosion behavior also seemed to relate to the sluggish diffusion. Dissolution of Cu was dominant in the competitive process of formation of the passive film. Sluggish diffusion can inhibit the migration of Cu, reduce the defects in the passive film, and inhibit the formation of cation vacancies, thereby improving corrosion behavior [35]. In this study, due to the segregation of Cu and Cu-rich areas exposed to the surface, the effect of slow diffusion was not obvious. Only in AlCrTiV_0.5_Cu_0.2_ HEA, in which the segregation of Cu structure was lower, was the corrosion resistance reduced slowly. When the content of Cu is higher and the degree of segregation is greater, the high-entropy effect has little effect, and the corrosion resistance is significantly reduced.

## 4. Conclusions

(1)In a solution of 0.6 M NaCl, the electrochemical behavior of AlCrTiV_0.5_Cu*_x_* lightweight HEAs changes depending on the proportion of Cu. While AlCrTiV_0.5_ has the best corrosion resistance and the lowest self-corrosion current density at 0.131 μA/cm^2^, AlCrTiV_0.5_Cu has the greatest self-corrosion current density at 2.778 μA/cm^2^ and the poorest corrosion resistance. AlCrTiV_0.5_ has the lowest self-corrosion current density and the best corrosion resistance.(2)When Cu is added to the HEAs, it is polarised between the dendrites, forming a Cu-rich phase. A higher Cu content leads to more pronounced polarization. The segregation of Cu lead to a large potential difference between the Cu-rich and Cu-poor phases that formed between and within the dendrites, making the area between the dendrites more susceptible to galvanic coupling corrosion.

## Figures and Tables

**Figure 1 materials-16-02922-f001:**
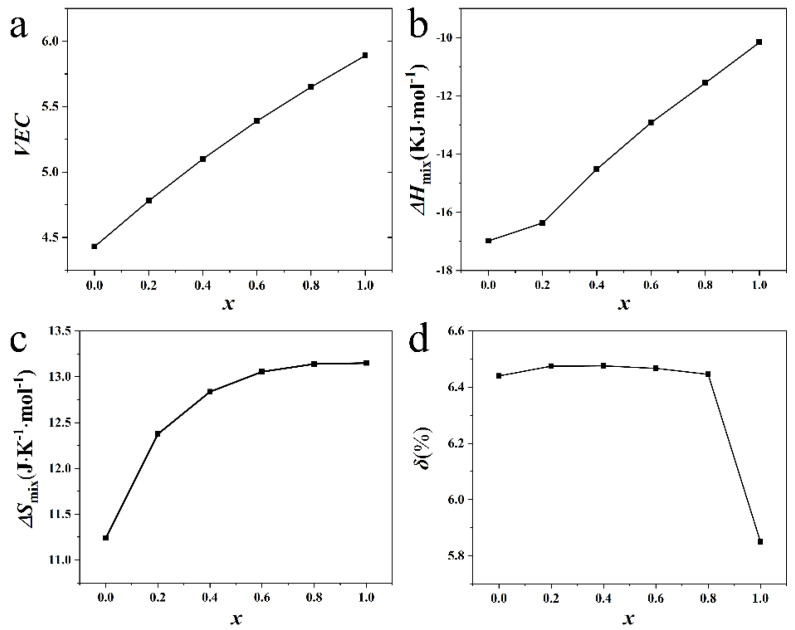
Variation in (**a**)VEC; (**b**) mixing enthalpy Δ*H*_mix_; (**c**) mixing entropy Δ*S*_mix_; (**d**) atomic size difference *δ* versus Cu content *x*(*x* = 0, 0.2, 0.4, 0.6, 0.8, 1) calculated for the AlCrTiV_0.5_Cu*_x_* lightweight HEAs.

**Figure 2 materials-16-02922-f002:**
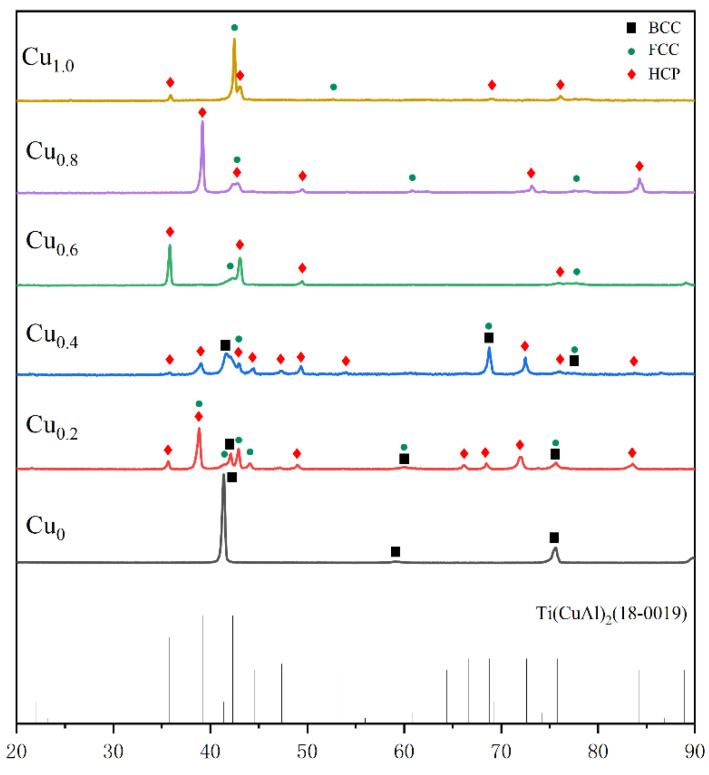
XRD patterns of AlCrTiV_0.5_Cu*_x_* lightweight HEAs.

**Figure 3 materials-16-02922-f003:**
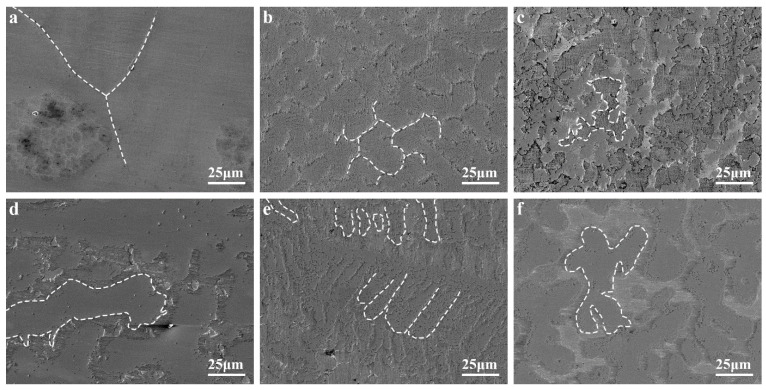
SEM micrographs of the AlCrTiV_0.5_Cu*_x_* lightweight HEAs. (**a**) *x* = 0; (**b**) *x* = 0.2; (**c**) *x* = 0.4; (**d**) *x* = 0.6; (**e**) *x* = 0.8; (**f**) *x* = 1.0.

**Figure 4 materials-16-02922-f004:**
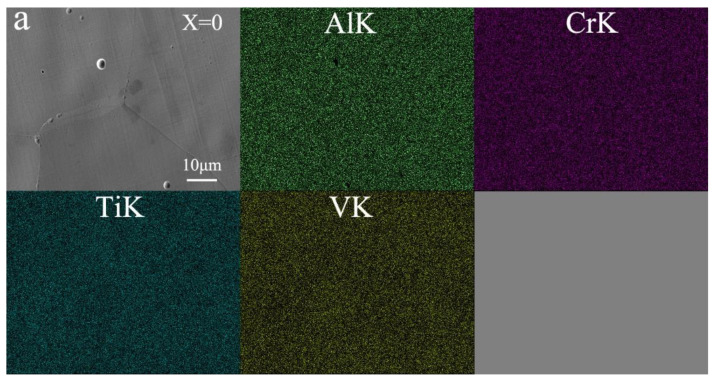
Elemental EDS mapping of AlCrTiV_0.5_Cu*_x_* lightweight HEAs. (**a**) x = 0; (**b**) x = 0.2; (**c**) x = 0.4; (**d**) x = 0.6; (**e**) x = 0.8; (**f**) x = 1.0.

**Figure 5 materials-16-02922-f005:**
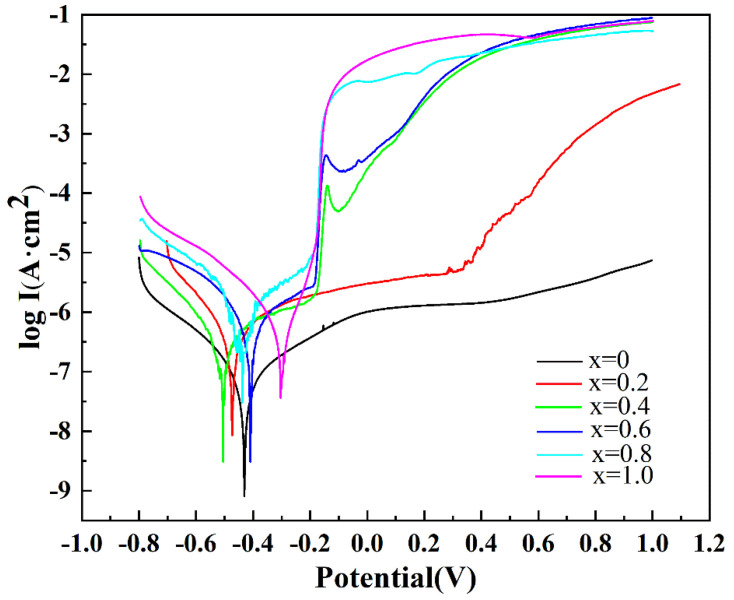
Potentiodynamic polarization curves of AlCrTiV_0.5_Cu*_x_* lightweight HEAs in 0.6 M NaCl solution.

**Figure 6 materials-16-02922-f006:**
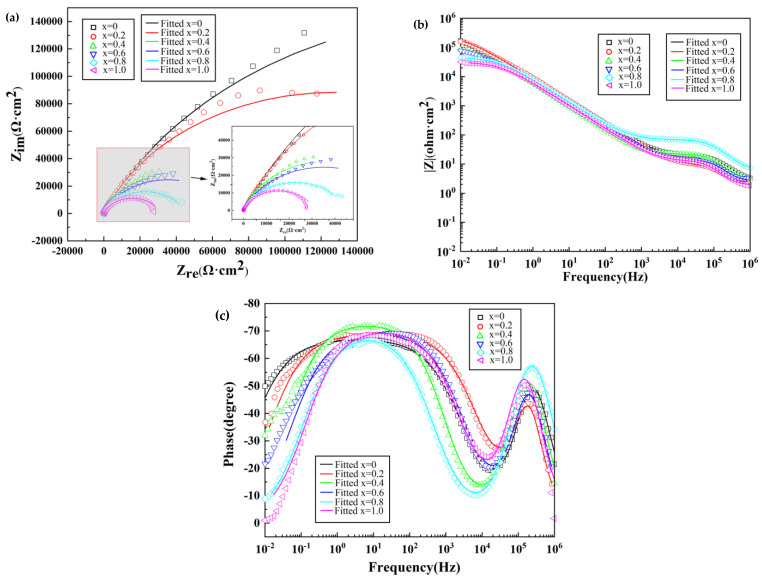
Fitted electrochemical AC impedance of AlCrTiV_0.5_Cu*_x_* lightweight HEAs in 0.6 M NaCl solution. (**a**) Nyquist plot; (**b**) Bode plot; (**c**) Phase angle plot.

**Figure 7 materials-16-02922-f007:**
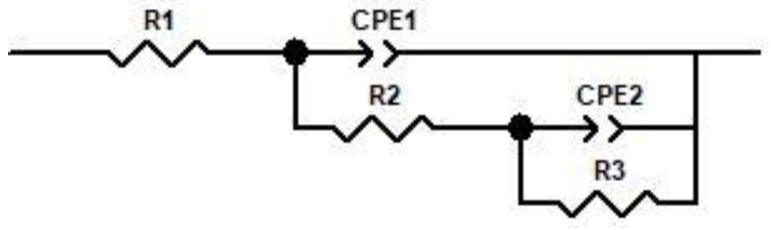
The equivalent electric circuit for impedance data.

**Figure 8 materials-16-02922-f008:**
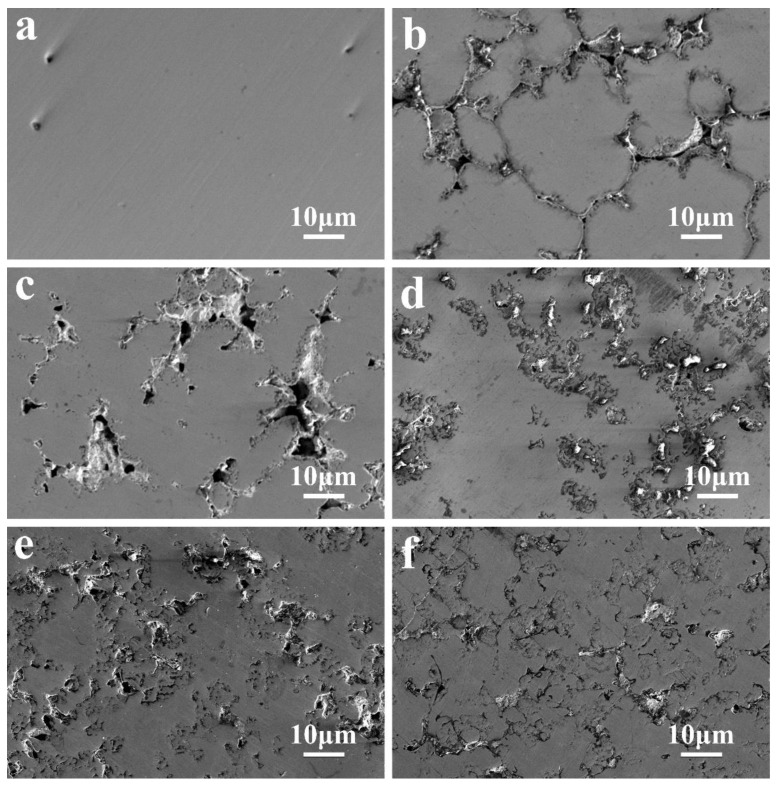
Surface morphology of AlCrTiV_0.5_Cu*_x_* lightweight HEAs after kinetic potential polarization tests in 0.6 M NaCl solution. (**a**) *x* = 0; (**b**) *x* = 0.2; (**c**) *x* = 0.4; (**d**) *x* = 0.6; (**e**) *x* = 0.8; (**f**) *x* = 1.0.

**Table 1 materials-16-02922-t001:** Electrochemical corrosion parameters of AlCrTiV_0.5_Cu*_x_* lightweight HEAs obtained from potentiodynamic polarization curve measured in 0.6 M NaCl solution.

Samples	*E*_corr_(V)	*I*_corr_(μA/cm^2^)
*x* = 0	−0.411	0.131
*x* = 0.2	−0.572	0.499
*x* = 0.4	−0.526	0.695
*x* = 0.6	−0.389	1.145
*x* = 0.8	−0.407	2.182
*x* = 1.0	−0.239	2.778
304 L	−0.415	4.7

**Table 2 materials-16-02922-t002:** Impedance fitting parameters of AlCrTiV_0.5_Cu*_x_* lightweight HEAs.

Samples	R_1_(Ω·cm^2^)	CPE_1_-T (μF·cm^−2^)	CPE_1_-P	R_2_(Ω·cm^2^)	CPE_2_-T (μF·cm^−2^)	CPE_2_-P	R_3_(Ω·cm^2^)
*x* = 0	3.183	0.0383	0.967	15.09	34.954	0.747	437,920
*x* = 0.2	2.444	0.0359	0.962	6.416	28.217	0.770	255,620
*x* = 0.4	3.106	0.0659	0.994	18.17	38.641	0.817	75,311
*x* = 0.6	2.818	0.0462	0.959	10.8	33.676	0.775	70,653
*x* = 0.8	5.808	0.0344	0.977	59.19	30.418	0.782	45,098
*x* = 1.0	1.934	0.0862	0.871	10.56	37.656	0.790	31,276

## Data Availability

The data presented in this work are available on request from the corresponding authors.

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
