# Peer review of "A Lightweight AlCrTiV_0.5_Cu*_x_* High-Entropy Alloy with Excellent Corrosion Resistance"

_materials, 2023, doi:10.3390/ma16072922_

Round 1
Reviewer 1 Report
The manuscript presents a relative interest in lightweight high-entropy alloys (HEAs). The manuscript is excellent in investigating electrochemical corrosion behavior. Still, based on discipline, novelty, and general significance, the manuscript lacks them. In addition to the usual criteria for publication in scholarly journals, this manuscript has not indicated what the authors have proposed, developed, or discovered. At the same time, it is more likely to be a research report but not a journal article. Regarding these, it is not recommended to be published in this journal. The results could be significant from a technological or laboratory point of view, but from a fundamental point of view, there is no real scientific contribution.
Author Response
First of all, we would like to thank the reviewers for reviewing our manuscript very carefully and putting forth many constructive comments. We have revised the manuscript based on these valuable comments and improved the quality of the manuscript. Changes are indicated in red font in the revised manuscript. We believe that our revisions and responses have addressed all the comments and concerns put forward by the reviewers and substantially strengthened our work. Our responses to the comments are as follows.
Reviewer: 1
Comments to the Author
The manuscript presents a relative interest in lightweight high-entropy alloys (HEAs). The manuscript is excellent in investigating electrochemical corrosion behavior. Still, based on discipline, novelty, and general significance, the manuscript lacks them. In addition to the usual criteria for publication in scholarly journals, this manuscript has not indicated what the authors have proposed, developed, or discovered. At the same time, it is more likely to be a research report but not a journal article. Regarding these, it is not recommended to be published in this journal. The results could be significant from a technological or laboratory point of view, but from a fundamental point of view, there is no real scientific contribution.
The manuscript presents a relative interest in lightweight high-entropy alloys (HEAs). The manuscript is excellent in investigating electrochemical corrosion behavior.
Response:
The authors would like to thank the Reviewer once again for the very careful reading of our manuscript, constructive comments, patience and overall effort that you put during the review process, which increased the scientific impact. The manuscript has modified and supplemented the contents which are not clear enough. The specific contents are as follows:
- Still, based on discipline, novelty, and general significance, the manuscript lacks them.
The innovation of this paper is mainly embodied in the following two points.
(1) A design strategy of lightweight HEAs is verified: the lightweight HEA system and alloying elements are selected, and the structure composition is predicted by calculating.
The results show that it is feasible to select metal elements as alloying elements in the lightweight HEA system and the phase structure can be predicted through the solid-solution phase formation rules of HEAs.
(2) The effect of Cu on the corrosion resistance of AlCrTiV0.5Cux lightweight HEA is studied. The results show that Cu decreases the corrosion resistance of AlCrTiV0.5Cux. Due to the high entropy effect, a small amount of Cu (x≤0.2) has little effect on the corrosion resistance of AlCrTiV0.5Cux lightweight HEA.
These two points are elaborated below:
The element in some HEA systems generally have a high density. The heavy component increases the energy consumption of the dynamic system. This is not conducive to high entropy alloy as a structural component, applied in automotive, transportation, aerospace and other fields. Considering the application environment of structural components, it is necessary to develop lightweight alloys with good corrosion resistance. In addition, the cost-effective also need to be considered from the economic point of view. At present, Light components, such as Al, Ti, Mg, and Li, are mainly added into the lightweight HEA. AlTiVCr lightweight HEA shows excellent corrosion resistance, which can inhibit pitting initiation more effectively than pure aluminum and 304 stainless steel. At the same time, it has been reported that the addition of Cu is beneficial to the improvement of corrosion resistance of some alloys. In this paper, cost-effective HEA with high corrosion resistance is obtained by reducing the content of V element and adding Cu element. However, Cu has a positive binary enthalpy of mixing and relatively high valence electron concentration (VEC) with other constitutional elements. The effect of Cu addition on the microstructure of high entropy alloy is observed. By calculating VEC, mixing enthalpy ΔHmix, mixing entropy ΔSmix, atomic size difference δ, the microstructure change of high entropy alloy after adding Cu is predicted. The change of corrosion resistance of lightweight high entropy alloy after adding Cu was studied in corrosion experiment, and the corrosion mechanism was discussed.
2.In addition to the usual criteria for publication in scholarly journals, this manuscript has not indicated what the authors have proposed, developed, or discovered.
Response:
(1) In the paper, the idea of developing a new type of lightweight HEA is proposed. Firstly, the lightweight HEA system with good corrosion resistance and alloying elements with required performance are selected. The change of microstructure is predicted by the solid-solution phase formation rules of HEAs. This method ensures the designed high entropy alloy has lower density, and it is easy to determine the effect of alloying elements on corrosion resistance by controlling variables.
(2) AlCrTiV0.5Cux lightweight HEA has been developed in this paper. The HEA still has a good corrosion resistance when Cu content is low.
(3) This paper discovered that the corrosion resistance of AlCrTiV0.5Cux lightweight HEA decreases with the addition of Cu due to the segregation of Cu. The addition of Cu increases the mixing entropy of the system, the effect of the high-entropy effect on the corrosion resistance of the alloys should be considered. AlCrTiV0.5Cux lightweight HEA still has good corrosion resistance when the amount of content is low(x≤0.2).
3.At the same time, it is more likely to be a research report but not a journal article. Regarding these, it is not recommended to be published in this journal. The results could be significant from a technological or laboratory point of view, but from a fundamental point of view, there is no real scientific contribution.
Response: The discussion sections have been added to this article, focusing on the following two areas.
(1) The analysis of composition design is modified and marked red in section 3.1, page 7.
(2) The analysis of corrosion mechanism is modified and marked red in section 3.3, page 13. The corrosion resistance of AlCrTiV0.5Cux lightweight HEA decreases with the addition of Cu due to the segregation of Cu. However, the high entropy effect can not be neglected. Sluggish diffusion can inhibit migration of Cu, reduced the defects of the passive film, inhibit formation of cation vacancies , thereby improving the corrosion behavior. That is why the addition of Cu has little effect on the corrosion resistance of AlCrTiV0.5Cux lightweight HEA when x ≤ 0.2.

Reviewer 2 Report
In this work, the authors studied the corrosion behavior of AlCrTiV0.5Cux high entropy alloys. It has been found that the corrosion resistance decreases with increasing Cu concentration. The work is interesting and novel. It is publishable subject to revision.
1.The authors claim that the studied alloys are lightweight. Al and Ti are light elements; however, the remainder are not. You should present the alloy density to confirm the statement. The density should be less than 5 g cm-3 for a lightweight alloy.
2.The authors should use subscripts and superscripts consistently. AlCrTiV0.5Vx should be rewritten AlCrTiV0.5Cux, g cm-3 should be g cm-3, etc.
3.You should pick another abbreviation for high performance additives or leave the term without abbreviating it (line 52). HEAs are used for high-entropy alloys.
4.What was the concentration of the KCl solution in the Ag/AgCl electrode? Was it a saturated solution? You need to specify it, as it determines the reference potential.
5.The suggested phase constitution (HCP, FCC, lines 157-158) must be confirmed by XRD. It is not enough to present the EDS data.
6.Have you measured an open-circuit potential before the polarization? If yes, present the data.
7.You claim that Cu was segregated at grain boundaries (line 174). Do you have an experimental verification (TEM/EDS data from the grain boundaries)?
Author Response
Responses to the reviewers
First of all, we would like to thank the reviewers for reviewing our manuscript very carefully and putting forth many constructive comments. We have revised the manuscript based on these valuable comments and improved the quality of the manuscript. Changes are indicated in red font in the revised manuscript. We believe that our revisions and responses have addressed all the comments and concerns put forward by the reviewers and substantially strengthened our work. Our responses to the comments are as follows.
Reviewer: 2
Comments to the Author
In this work, the authors studied the corrosion behavior of AlCrTiV0.5Cux high entropy alloys. It has been found that the corrosion resistance decreases with increasing Cu concentration. The work is interesting and novel. It is publishable subject to revision.
1.The authors claim that the studied alloys are lightweight. Al and Ti are light elements; however, the remainder are not. You should present the alloy density to confirm the statement. The density should be less than 5 g cm-3 for a lightweight alloy.
Response: We would like to thank the reviewer for the constructive comments. At present, the research on lightweight high entropy alloys is still in the preliminary stage, and the definition of lightweight high entropy alloys has not been clear. Based on the works of Senkov ON and Liaw PK, the high entropy alloys of density ρ<7g/cm3 are lightweight high entropy alloys.( Senkov ON et al. Acta Materialia.2013,61,1545; Liaw PK et al. Entropy 2016, 18,9, 333)
The alloy density is measured by the Archimedes' principle as follows:
The density of AlCrTiV0.5Cux HEAs
X value |
density(g/cm3) |
X=0 |
4.91 |
X=0.2 |
5.00 |
X=0.4 |
5.19 |
X=0.6 |
5.37 |
X=0.8 |
5.48 |
X=1.0 |
5.63 |
2.The authors should use subscripts and superscripts consistently. AlCrTiV0.5Vx should be rewritten AlCrTiV0.5Cux, g cm-3 should be g cm-3, etc.
Response: Thank you for your reminder. I have checked the subscripts and superscripts carefully, and indicated in red font in the revised manuscript.
3.You should pick another abbreviation for high performance additives or leave the term without abbreviating it (line 52). HEAs are used for high-entropy alloys.
Response: Thank you for your comments. I have deleted the abbreviation.
4.What was the concentration of the KCl solution in the Ag/AgCl electrode? Was it a saturated solution? You need to specify it, as it determines the reference potential.
Response: Thank you for your comments. The KCl solution in the Ag/AgCl electrode is 3.5 mol/L, which is a saturated solution.
5.The suggested phase constitution (HCP, FCC, lines 157-158) must be confirmed by XRD. It is not enough to present the EDS data.
Response: We would like to thank the reviewer for the constructive comments. The figure below is the XRD diagram of the sample. It can be seen from the figure that the sample of Cu0 shows a single BCC phase. After adding the Cu element, the samples of Cu0.2 and Cu0.4 consist of BCC phase, FCC phase and HCP. HCP phase is Ti(CuAl)2. With the further improvement of the Cu content, the samples of Cu0.6, Cu0.8, and Cu1.0 consist of FCC phase and HCP phase.
6.Have you measured an open-circuit potential before the polarization? If yes, present the data.
Response: Thank you for your comments. We measured the opening potential before the experiment and tested the impedance spectrum and polarization curve after the stability, and ensured that its fluctuation range was ± 0.05V, but it did not record.
7.You claim that Cu was segregated at grain boundaries (line 174). Do you have an experimental verification (TEM/EDS data from the grain boundaries)?
Response: We would like to thank the reviewer for the constructive comments.
The figure below shows the EDS map of Cu0.8, where the Spot 1 is located at the interface of the rich Cu phase and the rich Cr phase. The specific content is shown in the table below.

Reviewer 3 Report
I have the following comments in the submitted article.
Abstract-extend with concrete results
line 98 - for what reason was the concentration of 0.6 M NaCl determined?
line 100 - for what reason was the cross-section of the sample 0.5 cm2, when the cast samples were cubes with a side length of 5 mm?
Please insert the basic chemical composition without Cu.
Fig. 1 Enlarge the images and also the text on the axes. (very small font - illegible)
Fig. 2 Enlarge the images + line 142 (I observe the grain boundaries only in Fig. 2 and I will ask for either a graphic evaluation of the grains or a graphic indication of these boundaries on individual images. Alternatively, you can be inspired by, for example, the work from MDPI https://www.mdpi.com/ 1996-1944/15/5/1753 )
Fig. 3 Enlarge the images to the full width of the page + where the grain pots are shown (what do you mean by boundaries? primary grain? but phase boundary?)???
Fig. 4 What causes the fluctuation of the curve x0.6; x0.4? approximately at points -0.1 and -4
Fig. 5 Enlarge images and descriptions on axes + legend
Fig. 7 Enlarge images
Author Response
Responses to the reviewers
First of all, we would like to thank the reviewers for reviewing our manuscript very carefully and putting forth many constructive comments. We have revised the manuscript based on these valuable comments and improved the quality of the manuscript. Changes are indicated in red font in the revised manuscript. We believe that our revisions and responses have addressed all the comments and concerns put forward by the reviewers and substantially strengthened our work. Our responses to the comments are as follows.
Reviewer: 3
Comments to the Author
I have the following comments in the submitted article.
Abstract-extend with concrete results
line 98 - for what reason was the concentration of 0.6 M NaCl determined?
Response: We would like to thank the reviewer for the constructive comments.
0.6 M NaCl is 3.5wt%. It is the most common solution for studying alloy corrosion in a neutral environment.
line 100 - for what reason was the cross-section of the sample 0.5 cm2, when the cast samples were cubes with a side length of 5 mm?
Response: We would like to thank the reviewer for the constructive comments.
this is our error. We have revised to 25 mm2.
Please insert the basic chemical composition without Cu.
Response: Thank you for your comments. The basic chemical composition without Cu is AlCrTiV0.5.we have inserted it.
Fig. 1 Enlarge the images and also the text on the axes. (very small font - illegible)
Response: Thank you for your comments. We have enlarged the text on the axis, and reinsert it into the article. As shown below.
Fig. 2 Enlarge the images + line 142 (I observe the grain boundaries only in Fig. 2 and I will ask for either a graphic evaluation of the grains or a graphic indication of these boundaries on individual images. Alternatively, you can be inspired by, for example, the work from MDPI https://www.mdpi.com/ 1996-1944/15/5/1753 )
Response: And thank you for your suggestion. I have referred to this paper and enlarge the images as follows:
Fig. 3 Enlarge the images to the full width of the page + where the grain pots are shown (what do you mean by boundaries? primary grain? but phase boundary?)???
Response: Thank you for your comments. I have enlarged the picture to the full width of the page. As shown below. The grain structure of this alloy is a dendritic crystal, the boundary here refers to the boundary between the rich Cu phase and the rich Cr phase.
Fig. 4 What causes the fluctuation of the curve x0.6; x0.4? approximately at points -0.1 and -4
Response: Thank you for your comments. -0.4V at the polarization potential of Figure 4, the fluctuation of the "V" shape is the current peak that involves the transformation of activation and passivation, that is, the transition from the cathode reaction to anode reaction; when the potential increases to -0.2V, the current increases rapidly, the reason is, the passivation film formed by the initial passivation for the first time is not enough to inhibit the anode solubility process caused by the elevation of the polarization potential. Subsequently, the curve decreased slightly at -0.1V, which indicates that the sample generates a new passivation film under the potential.
Fig. 5 Enlarge images and descriptions on axes + legend
Response: Thank you for your comments. I have enlarged images and descriptions on axes and legend as below.
Fig. 7 Enlarge images
Response: Thank you for your comments. I have enlarged images as below.

Round 2
Reviewer 1 Report
The manuscript has been modified enough to be published at Materials.
Author Response
Thank you again for your comments
Reviewer 2 Report
Authors answered my comments and improved their paper. However, the XRD patterns from the cover letter should be included in the paper itself. After their inclusion, the manuscript can be accepted for publication.
Author Response
Authors answered my comments and improved their paper. However, the XRD patterns from the cover letter should be included in the paper itself. After their inclusion, the manuscript can be accepted for publication.
Response : Thank you for your comments, we added XRD patterns(Figure 2) in the revised manuscript.
Reviewer 3 Report
I thank the authors of the article for incorporating my comments.
I still have these small comments and questions about the work.
Fig. 2 I don't see grain boundary markings in the revised part of the article and no recommended literature (even in the Responses to the reviewers you have indicated boundaries). I don't know if there was just some mistake when uploading the corrected article. In Fig. 2e I don't think the grain boundaries are well marked.
Fig. 3 I also do not see the enlarged image in the revised part of the article
Author Response
Fig. 2 I don't see grain boundary markings in the revised part of the article and no recommended literature (even in the Responses to the reviewers you have indicated boundaries). I don't know if there was just some mistake when uploading the corrected article. In Fig. 2e I don't think the grain boundaries are well marked.
Response : Thank you for your suggestion, I have re-corrected the image and replaced it in the text as shown in Fig.3 in revised manuscript.
Fig. 3 I also do not see the enlarged image in the revised part of the article
Response : I have enlarged the images. And thank you for your suggestion.
Round 3
Reviewer 3 Report
Overall, the article is prepared at a high-quality level, it offers the reader information about the influence of Cu on the corrosion resistance of Lightweight high-entropy alloys. I recommend the article for publication.